# Reduction of Aflatoxin B1 and Zearalenone Contents in Corn Using Power Ultrasound and Its Effects on Corn Quality

**DOI:** 10.3390/toxins14120834

**Published:** 2022-11-30

**Authors:** Yuanfang Liu, Yuanxiao Liu, Wenbo Zhao, Mengmeng Li, Na Liu, Ke Bian

**Affiliations:** 1Department of Chemistry, Zhengzhou Normal University, No. 6, Yingcai Street, Huiji District, Zhengzhou 450044, China; 2College of Grain and Oil Food, Henan University of Technology, No. 100, Lianhua Street, Gaoxin District, Zhengzhou 450001, China

**Keywords:** aflatoxin B1, zearalenone, power ultrasound, corn quality

## Abstract

The degradation of aflatoxin B1 (AFB1) and zearalenone (ZEA) is investigated using power ultrasound to identify suitable methods to reduce the mycotoxin content of corn. AFB1 and ZEA in corn are simultaneously degraded via power ultrasound; thus, this method has a significant effect on corn quality. The power intensity, solid-liquid ratio, and ultrasonic treatment modes significantly affect the degradation rates of AFB1 and ZEA. The dissolution of AFB1 and ZEA in water also facilitates their degradation. At the initial stage of ultrasonic treatment, power ultrasound promotes the dissolution of mycotoxins in water, whereupon they are partially oxidized by free radicals. With a treatment time of 10 min, the reduction rates decreased owing to the dissolution of combined-state mycotoxins. After ultrasonic treatment, the contents of the essential amino acids, the total number of amino acids, and the fatty acids in corn decreased; however, ΔH values decreased during starch gelatinization. In contrast, the amylose content and viscosity of corn significantly increased during gelatinization. Therefore, this method is potentially suitable for the reduction of AFB1 and ZEA contents in corn.

## 1. Introduction

Corn is one of the main cereal grains used in food and animal feed. Corn production accounts for approximately 35% of the total global grain output. Therefore, the safe storage and processing of corn and corn products are essential for the agricultural economy worldwide. Notably, corn is susceptible to contamination by mycotoxins, including aflatoxins, zearalenone, deoxynivalenol, and ochratoxins [1]. Moreover, the same corn plant may be contaminated by multiple mycotoxins, of which aflatoxin B1 (AFB1) and zearalenone (ZEA) are the most common. Several previous studies indicate that the type and amount of mycotoxin contamination in corn depends on the country and area in which it is grown [2,3,4]. Moreover, AFB1 and ZEA elicit great public health concerns due to their high prevalence and their teratogenic, carcinogenic, mutagenic, and immunosuppressive effects [5,6,7,8].

Numerous methods have been developed to reduce mycotoxin contamination in cereals, including physical, chemical, and biological methods [9,10,11]. Although many of these methods efficiently reduce mycotoxins in corn, only a few have been used industrially because they also reduce the number of nutrients in cereals and adversely affect their quality. Moreover, these methods may require expensive instruments or cause secondary pollution [12,13]; therefore, new methods of degrading mycotoxins in corn are essential. Ultrasonic waves are mechanical waves with frequencies above 20 kHz. Power ultrasound is a type of ultrasonic wave with frequencies ranging from 20 to 100 kHz, and it has been applied in enzymatic inactivation, sterilization, protein modification, and the freeze-drying of foods owing to its high energy [14,15,16]. Power ultrasound is known to change the structure of some low-molecular-weight compounds owing to the cavitation effect, which has been exploited to degrade pesticide residues in fruits and vegetables [17]. Based on these earlier studies, we inferred that power ultrasound might be suitable for the degradation of mycotoxins in corn.

In this study, we used power ultrasound to reduce the levels of AFB1 and ZEA in corn. In addition, we comprehensively studied the effects of power intensity, solid-liquid ratio, treatment time, and modes of power ultrasound on the degradation rates of AFB1 and ZEA, along with the effects of power ultrasound on corn quality. This study provides a theoretical basis for the industrial application of this technology.

## 2. Results and Discussion

### 2.1. Degradation of AFB1

The reduction of the AFB1 content in corn using power ultrasound and the effects of power intensity, solid-liquid ratio, and treatment modes on the degradation rates of AFB1 were comprehensively examined.

#### 2.1.1. Power Intensity

Figure 1 shows the effects of three different power intensities on the degradation rates of AFB1 in water and corn over time.

The degradation rates of AFB1 did not change significantly after ultrasonic treatment for 2–8 min; however, a significant increase in the degradation rate was observed with a treatment time of 10 min (*p* < 0.05). The increase in the treatment time from 2 to 10 min promoted the dissolution of AFB1 in water; however, the degradation rate significantly decreased with a further increase in the treatment time from 10 min to 12 min (*p* < 0.05). The transfer of AFB1 from water to corn powder and the precipitation of AFB1 may explain this observation [18]. Another reason for the result is that AFB1 exists in two forms in corn: the free form and the masked form [18]. After long periods of ultrasound treatment, the masked AFB1 in corn may transform into free AFB1, which can be detected accurately. Moreover, the variations in the reduction rates at different power intensities were similar, and the degradation rate of AFB1 at the power intensity of 1.65 W/cm^3^ was the highest at 16.02%. This might be because cavitation bubbles are generated more easily at higher power intensities, and the bubbles burst more fiercely when the power intensity is increased from 0 to 1.65 W/cm^3^ [19]. However, the spread of ultrasonic waves may have been restrained at the power intensity of 2.2 W/cm^3^ owing to the large number of cavitation bubbles [20]. At the power intensities of 1.1, 1.65, and 2.2 W/cm^3^, the degradation rate of AFB1 in the soaking water did not change significantly when the ultrasonic treatment time was 2–10 min but increased significantly at 12 min (*p* < 0.05). The increase in the AFB1 content in water was derived from the corn powder; the AFB1 dissolved in water was degraded by ultrasonic power more effectively than the AFB1 in corn [21]. However, a high amount of AFB1 was transferred from corn to water with an increase in treatment time, thereby decreasing the degradation rate of AFB1 in water.

#### 2.1.2. Solid-Liquid Ratio

Figure 2 shows the effects of three different solid-liquid ratios on the degradation rates of AFB1 in corn and soaking water.

The degradation rates of AFB1 were largely unaffected during treatment times between 2 and 8 min, regardless of the solid-liquid ratio. However, the degradation rate increased significantly when the treatment time was 10 min and significantly decreased upon increasing the treatment time from 10 to 12 min (*p* < 0.05). A treatment time of 10 min resulted in the highest AFB1 degradation rate. However, this rate significantly decreased with an increase in the solid-liquid ratio from 10/100 to 20/100 (*p* < 0.05), which was mainly caused by the reduction in the ultrasonic effects [14]. The degradation rate of AFB1 in water increased with the treatment time, and the highest rate was observed with a solid-liquid ratio of 10/100 and a treatment time of 12 min (*p* < 0.05). These results demonstrate the beneficial effect of ultrasonic treatment on the dissolution of AFB1.

#### 2.1.3. Treatment Modes

The effects of two different treatment modes (continuous and pulsed ultrasonic treatments) on the rate of AFB1 degradation rate are shown in Figure 3.

As the power intensity increased, the degradation rates initially increased and then decreased. At power intensities of 0.55 W/cm^3^ and 1.1 W/cm^3^, the degradation rates observed during the continuous ultrasonic treatment were significantly higher than those observed during the pulsed ultrasonic treatment (*p* < 0.05). Conversely, opposite results were observed at 1.65 W/cm^3^ and 2.2 W/cm^3^. A power intensity of 1.65 W/cm^3^ resulted in the highest degradation rate of AFB1. Moreover, the degradation rate during pulsed ultrasonic treatment was higher than that during continuous ultrasonic treatment. When the power intensity density was 1.65 W/cm^3^, the degradation rate of AFB1 was up to 21.2% after 10 min of pulsed ultrasonic treatment. The above results demonstrate that the correlation between power intensity and the treatment modes during the ultrasonic treatments significantly affected the degradation rates. At a power intensity of 1.65 W/cm^3^, the oxidization effects of ultrasound could be enhanced by pulsed ultrasonic treatment.

### 2.2. Degradation of ZEA

The reduction of the ZEA content in corn using power ultrasound and the effects of power intensity, solid-liquid ratio, and treatment modes on the degradation rates of ZEA were studied.

#### 2.2.1. Power Intensity

Figure 4 shows the effect of three power intensities on the degradation rates of ZEA in corn and water.

With increasing treatment time, the degradation rates initially increased and then decreased. With a power intensity of 1.1 W/cm^3^, the degradation rate of ZEA was the highest at the treatment time of 8 min (*p* < 0.05). However, at power intensities of 1.65 and 2.2 W/cm^3^, the degradation rate was the highest at 6 min (*p* < 0.05). These results indicate that the increase in the treatment time from 6 min to 8 min promoted the dissolution of ZEA in water. Conversely, the degradation rate significantly decreased at treatment times greater than 8 min owing to the dissolution of combined ZEA (*p* < 0.05) [18]. The variation in the ZEA degradation rate with increasing treatment time was similar to that observed with increasing power intensity, and the highest reduction rate (34.6%) was observed when the power intensity was 1.1 W/cm^3^ and the treatment time was 8 min. Moreover, the ZEA degradation rate in water significantly increased with an increase in the treatment time from 2 to 8 min (*p* < 0.05). However, it significantly decreased with an increase in the treatment time from 8 to 12 min, which may be explained by the dissolution of ZEA in water.

#### 2.2.2. Solid-Liquid Ratio

Figure 5 shows the effect of three different solid-liquid ratios on the degradation rate of ZEA in corn and water.

With increasing treatment time, the ZEA degradation rate first increased and then decreased (*p* < 0.05). The variation in these rates was similar at each solid-liquid ratio. The highest ZEA degradation rate was observed with a treatment time of 6 min and a solid-liquid ratio of 10/100; however, the degradation rate decreased significantly with an increasing solid-liquid ratio (*p* < 0.05). Increasing the solid-liquid ratio is known to weaken ultrasonic effects [14], which may explain these observations. The degradation rate of ZEA in water also first increased and then decreased. The degradation rate in water was the highest at a treatment time of 6 min and a solid-liquid ratio of 20/100 (*p* < 0.05).

#### 2.2.3. Treatment Modes

Figure 6 shows the effects of two different treatment modes (continuous and pulsed ultrasonic treatments) on the degradation rate of ZEA.

The ZEA degradation rate exhibited different tendencies under different treatment modes. During continuous ultrasonic treatment, the degradation rate significantly increased upon increasing the power intensity from 0.55 W/cm^3^ to 1.1 W/cm^3^ and significantly decreased upon increasing the power intensity from 2.2 W/cm^3^ to 2.75 W/cm^3^ (*p* < 0.05). However, the degradation rate did not change significantly at power intensities ranging from 1.1 W/cm^3^ to 2.2 W/cm^3^ (*p* < 0.05). Conversely, during pulsed ultrasonic treatment, the degradation rate remained largely unaffected. At the power intensities of 0.55 W/cm^3^ to 2.75 W/cm^3^, the degradation rates observed during continuous ultrasonic treatment were lower than those observed during pulsed ultrasonic treatment. However, the opposite result was observed at power intensities ranging from 1.1 W/cm^3^ to 2.2 W/cm^3^.

### 2.3. Amino Acid Content

The changes in the amino acid content of corn after ultrasonic treatment are illustrated in Table 1. After the ultrasonic treatment, the contents of essential and total amino acids decreased; meanwhile, the ratio of essential to total amino acids did not change significantly (*p* < 0.05). The reduction in the amino acid content may be caused by the oxidation of the side chains of amino acids by the free radicals produced under ultrasonic treatment.

### 2.4. Fatty-Acid Value

The fatty-acid value is an important index that reflects the rancidity of cereals during storage. The rancidity of fatty acids is higher than the quality change velocities of starch and protein owing to oxidation and hydrolysis reactions [22]. The effect of ultrasonic treatment on the fatty acid values of corn is shown in Figure 7.

The fatty-acid values significantly decreased with increasing treatment time, which may be attributed to the partial dissolution of fatty acids under ultrasonic cavitation and the degradation of short-chain fatty acids.

### 2.5. Gelatinization Properties

The gelatinization properties of starch, which is the most abundant component of corn, play an essential role in determining corn quality. Table 2 summarizes the changes in the gelatinization properties of corn starch observed at various ultrasonic treatment times. With an increase in the treatment time, the peak viscosity, hold viscosity, and final viscosity increased. This may be a result of the crystal structure of damaged starch granules, which promotes water absorption [23]. In contrast, the breakdown value, setback value, and gelatinization temperature remained unaffected by changes in treatment time.

### 2.6. Thermodynamic Properties

Starch granules absorb water during the gelatinization of corn starch, causing them to swell. This transforms the ordered structure of starch molecules into a disordered state along with heat exchange. Accordingly, the gelatinization properties of corn starch were analyzed using differential scanning calorimetry (DSC) (Table 3). The onset temperature (T_0_), peak temperature (T_p_), conclusion temperature (T_c_), and transition temperature (T_c_–T_0_) did not change significantly after ultrasonic treatment, whereas the gelatinization enthalpy (ΔH) significantly decreased after 12 min of ultrasonic treatment. The reduction in ΔH at 12 min is closely associated with the breaking of hydrogen bonds and the disordering of starch molecules [24,25,26].

### 2.7. Amylose Contents

Table 4 shows the changes in the amylose content of corn starch after the ultrasonic treatment of corn powder, which was observed to have increased from 29.80% to 32.93% as treatment time increased. The amylose content of corn after 12 min of ultrasonic treatment was significantly higher (*p* < 0.05) than that of the control owing to the rupture of the α-1,6-glycosidic bond during the ultrasonic treatment [25,27,28]. Moreover, the starch granules could have been damaged, as hydrogen bonds are readily broken during ultrasonic treatment [25]. The changes in the amylose content were closely related to the changes in the viscosity of the starch samples during gelatinization.

### 2.8. Scanning Electron Microscopy

The microstructures and surface morphologies of corn powder were observed using scanning electron microscopy (SEM) (Figure 8).

Native corn starch exhibited polygonal and spherical morphologies and was closely integrated with protein. After ultrasonic treatment, notches and cavities appeared on the starch granules. Moreover, as treatment time increased, the number of notches and cavities increased, and the integration of starch with protein became loosely ordered. Water diffused within the starch granules through the notches and cavities on their surfaces, thus damaging the starch granules [29]. Under the effect of ultrasonic treatment, the specific surface area of the starch granules significantly increased, thus promoting the contact of free radicals with the starch granules, causing the breakage of more glycosidic bonds. Accordingly, starch gelatinization increased the amylose content and decreased ΔH.

## 3. Conclusions

AFB1 and ZEA in corn can be simultaneously degraded using power ultrasound. When the power intensity density was 1.65 W/cm^3^, the degradation rate of AFB1 was 21.2% after 10 min of pulsed ultrasonic treatment. The highest ZEA degradation rate (34.6%) was observed at the power intensity of 1.1 W/cm^3^ after 8 min of treatment. The power intensity, solid-liquid ratio, and ultrasonic treatment modes significantly affected the degradation rates of AFB1 and ZEA. The dissolution of AFB1 and ZEA in water facilitated their degradation during the ultrasonic treatment of corn. During the initial stage of ultrasonic treatment, power ultrasound promoted the dissolution of mycotoxins, which subsequently dissolved in water and were partly oxidized by free radicals under the effect of power ultrasound. With a treatment time of 10 min, the degradation rates of mycotoxins decreased owing to the dissolution of combined-state mycotoxins. In addition, the ultrasonic treatment significantly affected corn quality. The content of essential amino acids, total amino acids, and fatty acids in corn significantly decreased after ultrasonic treatment. During starch gelatinization, ΔH values decreased, but the viscosity of corn starch and its amylose content significantly increased. Therefore, this treatment may be suitable for reducing the contents of AFB1 and ZEA in corn. Future studies in this area will determine the optimal parameters of this method. In particular, the interaction among power intensity, solid-liquid ratio, treatment modes, and treatment time during the ultrasonic treatment of corn should be further explored. In addition, although ultrasonic treatment reduced the AFB1 and ZEA contents in corn, it adversely affected corn quality. Hence, the parameters of this method must be optimized to maximize the degradation rates of mycotoxins while retaining corn quality.

## 4. Materials and Methods

### 4.1. Materials

Corn kernels with a moisture content of 14.2%, AFB1 (358.6 μg/kg), and ZEA (472.2 μg/kg) were obtained from Xinxiang, China. AFB1 (purity > 98%) and ZEA (purity > 98%) standards were purchased from J&K Scientific Co., Ltd. (Shanghai, China). Formic acid (purity > 98%) and ammonium acetate were obtained from PanReac (Barcelona, Spain). LC-MS grade acetonitrile and methanol were supplied by Thermo Fisher Scientific (Rockford, IL, USA). Water (18.2 MΩ cm at 25 °C) was collected from an ultrapure purification system (Millipore, MA, USA).

### 4.2. Ultrasonic Treatment of Corn 

Corn kernels were first treated with liquid nitrogen before being ground into a powder until they were able to pass through a 40-mesh sieve [21]. The sample solutions were prepared by mixing an appropriate amount of corn powder (10, 15, and 20 g) with pure water (100 mL). A 550 W (adjustable) ultrasonic processor (SFX550D, Branson Ultrasonic Co., Shanghai, China) with a 13 mm probe was used for sonication. The probe was inserted into the solution at a depth of 15 mm, and the treatment was carried out. The removal rate of mycotoxins in corn flour was determined under the following experimental conditions: solid–liquid ratio (10/100, 15/100, or 20/100); degradation time (2, 4, 6, 8, 10, or 12 min); duty cycle (50% or 100%); power intensity (1.1, 1.65, or 2.2 W/cm^3^). After ultrasonic treatment, the solutions were centrifuged at 4000 rpm for 20 min before the sediments in the solution were freeze-dried for 48 h. The dried corn powder was stored at −20 °C for future use.

### 4.3. Preparation of the Mixed Standard Solutions of AFB1 and ZEA

AFB1 and ZEA standards were dissolved in a mixture of water and methanol (water/methanol = 50/50, *v*/*v*). The solution was then diluted to seven different concentrations (M1–M7), as listed in Table 5.

### 4.4. Extraction of AFB1 and ZEA 

The water (2 mL) used in the ultrasonic treatment of corn was evaporated under nitrogen flow. The residue was redissolved in a methanol/water (50/50, *v*/*v*) mixture (2.0 mL) under vortex oscillation for 1 min. The redissolved solution was then centrifuged at 12,000 rpm for 10 min, and the supernatant was collected for further analysis [21].

After ultrasonic treatment, corn powder (2.0 g) was mixed with the extracting solution (10 mL, methanol/water/glacial acetic acid = 79/20/1, *v*/*v*/*v*) and subjected to ultrasonication for 20 min. The solution was then centrifuged at 4000 rpm for 20 min. The supernatant (0.5 mL) was transferred into a centrifuge tube and mixed with pure water (0.5 mL). The mixed solution was oscillated for 40 s and centrifuged at 12,000 rpm for 15 min, and the supernatant was collected for further analysis.

### 4.5. Determination of AFB1 and ZEA 

The concentrations of AFB1 and ZEA were determined using ultra-performance liquid chromatography (UPLC) combined with tandem mass spectrometry (UPLC-MS/MS) [30]. A C18 column (100 × 2.1 mm, particle size 1.7 mm, Thermo Fischer Scientific, Dreieich, Germany) was used for separation. The column temperature was set to 35 °C, and the flow rate and injection volume were 0.30 mL/min and 5 μL, respectively. The mobile phase initially contained 75% of A (water containing 5 mmol/L of ammonium acetate and 0.1% formic acid) and 25% of B (methanol). After 1.5 min, the mobile phase was changed to 95% A and maintained at 95% A from 1.5 min to 4.2 min. Finally, the mobile phase was changed to 75% A from 4.2 min to 4.3 min and maintained until the end of the experiment (6 min).

A Q-Orbitrap MS/MS instrument (Q-Exactive, Thermo Fisher Scientific, Germany) equipped with a heating electrospray ionization source was used for the simultaneous detection of AFB1 and ZEA using the full MS mode. The relevant parameters were set as follows: sheath gas flow rate: 25 arb; auxiliary gas flow rate: 5 arb; spray voltage: 3.0 kV; capillary temperature: 250 °C; auxiliary gas temperature: 300 °C; scan range: 150–600 m/z; resolution: 70,000.

### 4.6. Quantitative Analysis

Quantitative analysis of AFB1 and ZEA was achieved using the external standard method. The standard curve equations and confirmation results are listed in Table 6.

### 4.7. Amino Acid Content 

Corn powder (0.1 g) was dissolved in hydrochloric acid (6 mol/L, 8 mL) and bubbled with nitrogen gas, followed by hydrolysis at 120 °C for 2 h. Sodium hydroxide (10 mol/L, 4.8 mL) was then added to the previously mentioned solution [31]. The solution was filtered and diluted to 25 mL. The content of amino acids in the supernatant was determined using an Agilent-1200 HPLC system (Agilent Technologies, Waldbronn, Germany) equipped with a quaternary pump, online degasser, auto-sampler, and column heater–cooler. Separation was performed on an Agilent SB-C18 column (250 × 4.6 mm, particle size 5 mm; Agilent Technologies, Newport, DE, USA). Chromatographic analyses were carried out using gradient elution with mobile phase A, consisting of 0.05 mol/L sodium acetate buffer solution (pH = 6.50), and mobile phase B, consisting of an acetonitrile-water solution (*v*/*v* = 1:1). Gradient elution was initiated with 30% B, which was held for 5 min. Thereafter, it was increased linearly up to 100% B in 35.0 min. This composition was held for another 2.0 min before being reduced to 16% B in 7.0 min, followed by a re-equilibration time of 6.0 min, yielding a total run time of 55.0 min. The flow rate was set at 1.0 mL/min, and column temperature was maintained at 23 °C. The detection wavelength was 360 nm. Aliquots of 10 μL of the sample extract were injected into the chromatographic system.

### 4.8. Fatty-Acid Value 

The fatty acid values of corn were determined according to the standard method described in “GB/T 5510-2011 Inspection of grain and oils: Determination of fat acidity value of grain and oilseeds.” [32] Corn powder (5.0 g) was placed in a conical flask with 50.0 mL of petroleum ether for 10 min. The filtrate (25.0 mL) was mixed with 75% alcohol (75.0 mL) and 3–4 drops of phenolphthalein indicator and then titrated with KOH solution until the solution had a faint pink color. The fatty acid values of corn were obtained based on the amount of consumed KOH standard titrant.

### 4.9. Gelatinization Properties 

The gelatinization properties of corn were determined using a Rapid Visco Analyzer (Perten Instrument Co., Ltd., Stockholm, Sweden) according to the standard method described in AACC Method 76-21 [33]. Corn powder (3.50 g) and water (25.0 ± 0.1 mL) were transferred into the test canister. A stirrer was placed into the canister, and the blade was vigorously jogged through the sample up and down 10 times. The measurement cycle was then initiated, and the test proceeded and terminated automatically. The gelatinization properties of corn were noted from the recorded viscosity.

### 4.10. Thermodynamic Properties

The thermodynamic properties of corn were determined by DSC (TA Instruments Co., Ltd., Newcastle, DE, USA). The corn powder sample (3 mg) and pure water (9 µL) were added to a crucible, which was sealed and equilibrated at approximately 22 °C for 24 h. The thermodynamic properties were then determined in the scanning range of 30–100 °C at a heating rate of 10 °C/min. The gelatinization temperature (T_0_), peak temperature (T_p_), conclusion temperature (T_c_), and gelatinization enthalpy (ΔH) were recorded.

### 4.11. Amylose Content

The amylose content was determined according to the method reported by Yun et al. [34]. Corn powder (25 mg) was mixed with dimethyl sulfoxide (1 mL) and defatted using absolute ethyl alcohol. Amylopectin was removed by adding 20 mg of concanavalin A (Con A) into the defatted samples, followed by centrifugation. Amylose in the supernatant was hydrolyzed by adding glucose oxidase and peroxidase (GOPOD). Subsequently, 4-aminoantipyrine was added to the sample to induce a coloration reaction, and the absorbance was measured at a wavelength of 510 nm. The amylose content was calculated according to the following equation:$$\mathrm{Amylose}\;\mathrm{contents}\;(w/w)=\frac{A_{\mathrm{conA}}}{A_{\mathrm{TS}}}\times \frac{6.15}{9.2}\times \frac{100}{1}=\frac{A_{\mathrm{conA}}}{A_{\mathrm{TS}}}\times 66.8$$
where *A*_ConA_ is the absorbance of the supernatant, *A*_TS_ is the absorbance of the starch sample, and 6.15 and 9.2 are the dilution factors of Con A and starch, respectively.

### 4.12. SEM

Surface microstructures of the corn powder were observed using SEM at a magnification of 5000× and 2500×. For SEM, samples were dehydrated in 99.8% ethanol and sprinkled on a double-sided adhesive tape mounted on an aluminum stub. Thereafter, they were coated with a thin gold film using sputter coater Auto 108 (TED PELLA Inc., Redding, CA, USA) and observed in a Quanta microscope (250FEG, FEI Co., Hillsborough, OSU, USA) at an accelerating potential of 20 kV.

### 4.13. Statistical Analysis

Data are shown as the mean ± standard deviation of at least three parallel experiments. One-way analysis of variance was used to compare values from more than two different experimental groups, and a t-test was used to compare values between the two different experimental groups. *p* values less than 0.05, as determined by the Duncan multiple comparison method, were considered statistically significant. Statistical analysis was performed using the SPSS 16.0 software (SPSS Co., Ltd., Chicago, IL, USA). All diagrams were drawn with the Origin 8.5 software (Origin Lab Corporation, Northampton, MA, USA).

## Figures and Tables

**Figure 1 toxins-14-00834-f001:**
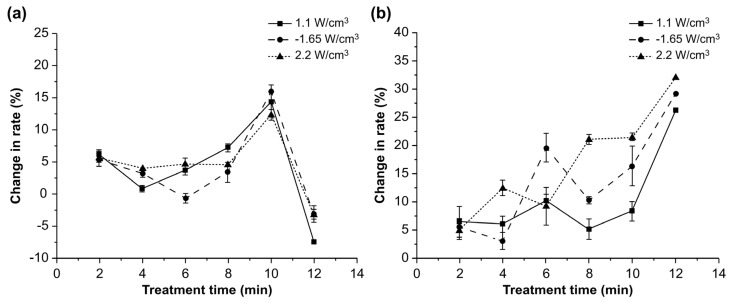
Effect of power intensity on the degradation rate of aflatoxin B1 in (**a**) corn and (**b**) water over time.

**Figure 2 toxins-14-00834-f002:**
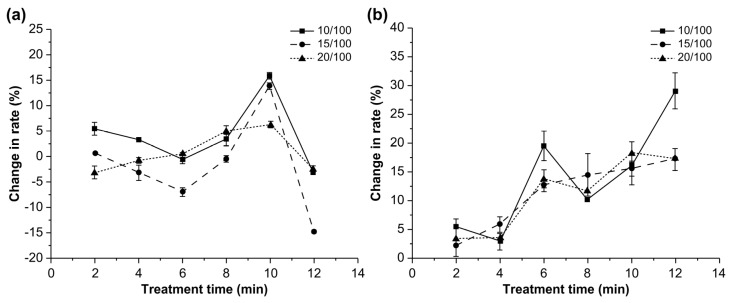
Effects of solid-liquid ratio on the degradation rate of aflatoxin B1 in (**a**) corn and (**b**) water.

**Figure 3 toxins-14-00834-f003:**
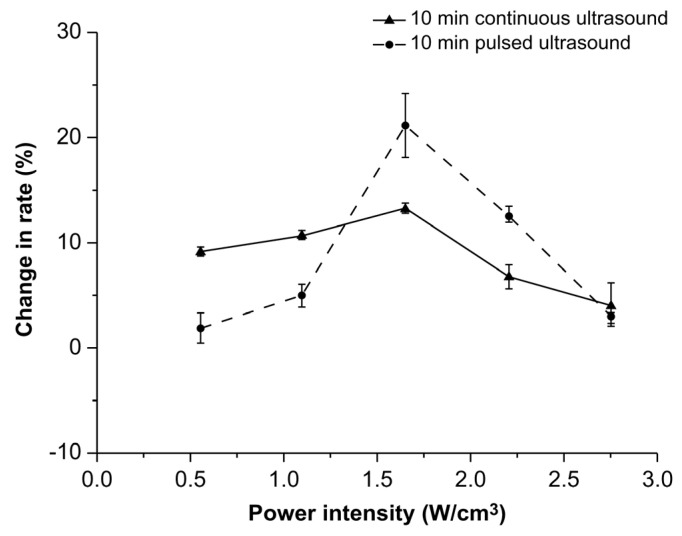
Effects of treatment modes on the degradation rate of aflatoxin B1 in corn.

**Figure 4 toxins-14-00834-f004:**
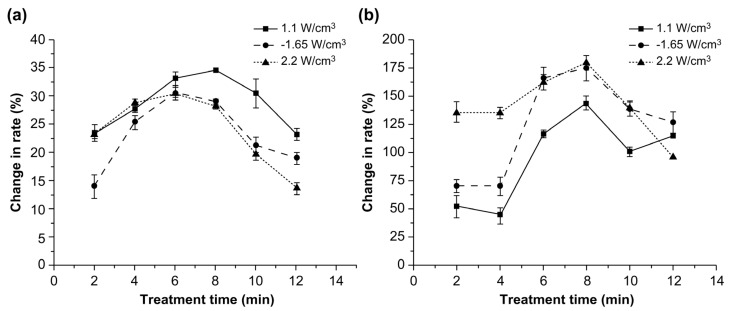
Effects of power intensity on the degradation rates of zearalenone in (**a**) corn and (**b**) water.

**Figure 5 toxins-14-00834-f005:**
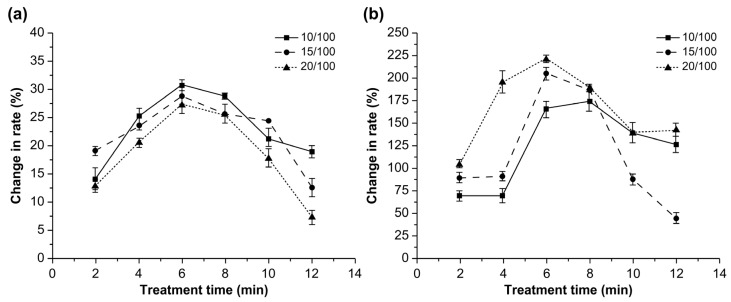
Effects of the solid-liquid ratio on the degradation rate of zearalenone in (**a**) corn and (**b**) water.

**Figure 6 toxins-14-00834-f006:**
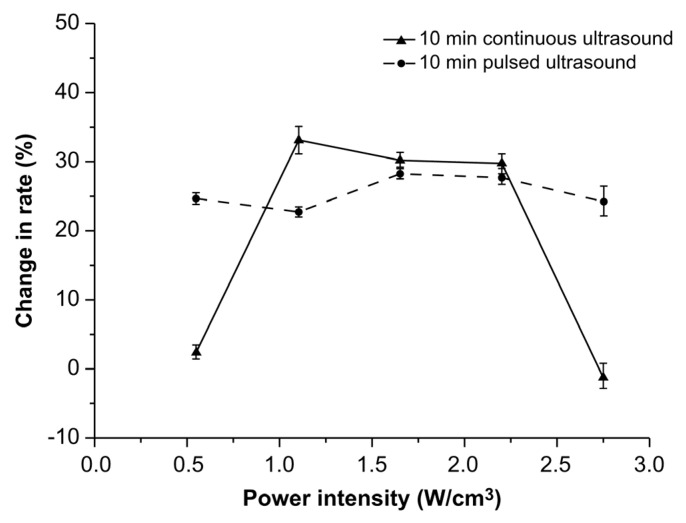
Effects of treatment modes on the degradation rate of zearalenone in corn.

**Figure 7 toxins-14-00834-f007:**
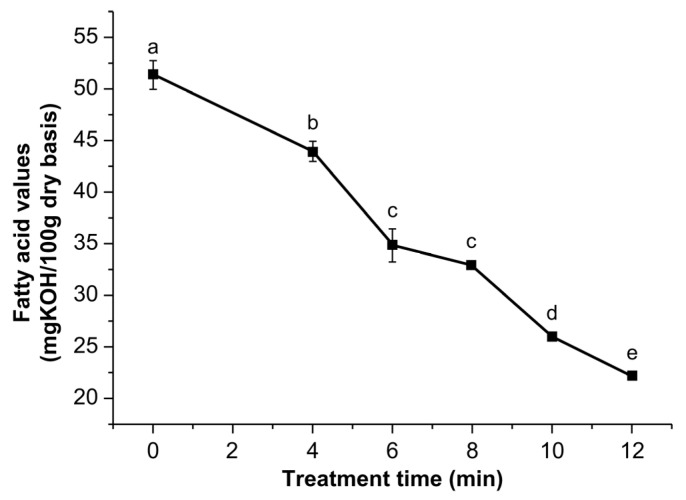
Changes in the fatty-acid values of corn treated by power ultrasound. Values with different letters (a, b, c, d, e) were significantly different (*p* < 0.05).

**Figure 8 toxins-14-00834-f008:**
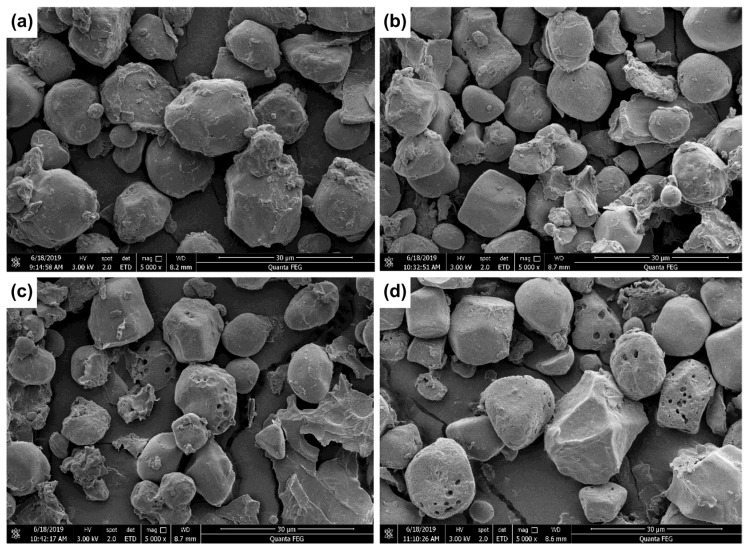
Scanning electron microscopy images of corn flour treated with power ultrasound at different times: (**a**) control sample (no-ultrasonic treatment) and corn flour samples treated by ultrasound for (**b**) 4 min, (**c**) 8 min, and (**d**) 12 min. Scale bar: 30 µm.

**Table 1 toxins-14-00834-t001:** Changes in the amino acid content in corn treated by power ultrasound.

	Control	6 min	10 min
Aspartic acid (Asp)	0.45 ± 0.02	0.42 ± 0.03	0.42 ± 0.00
Threonine (Thr)	0.26 ± 0.00	0.25 ± 0.01	0.25 ± 0.00
Serine (Ser)	0.36 ± 0.01	0.34 ± 0.01	0.35 ± 0.00
Glutamic acid (Glu)	1.81 ± 0.04	1.78 ± 0.04	1.77 ± 0.01
Glycine (Gly)	0.31 ± 0.00	0.29 ± 0.01	0.29 ± 0.00
Alanine (Ala)	0.58 ± 0.01	0.57 ± 0.02	0.56 ± 0.00
Cysteine (Cys)	0.06 ± 0.01	0.06 ± 0.01	0.06 ± 0.00
Valine (Val)	0.27 ± 0.00	0.25 ± 0.01	0.25 ± 0.01
Methionine (Met)	0.11 ± 0.02	0.11 ± 0.02	0.11 ± 0.01
Isoleucine (Ile)	0.24 ± 0.01	0.24 ± 0.01	0.23 ± 0.00
Leucine (Leu)	0.78 ± 0.03	0.81 ± 0.01	0.77 ± 0.01
Tyrosine (Tyr)	0.28 ± 0.00	0.25 ± 0.02	0.26 ± 0.00
Phenylalanine (Phe)	0.28 ± 0.00	0.28 ± 0.02	0.26 ± 0.01
Histidine (His)	0.29 ± 0.01	0.29 ± 0.01	0.29 ± 0.00
Lysine (Lys)	0.22 ± 0.01	0.20 ± 0.01	0.20 ± 0.00
Arginine (Arg)	0.30 ± 0.02	0.29 ± 0.01	0.29 ± 0.00
Proline (Pro)	0.62 ± 0.01	0.62 ± 0.00	0.62 ± 0.02
Essential amino acid (EAA)	2.15	2.14	2.08
Total amino acid (TAA)	7.23	7.05	6.99
EAA/TAA	0.30	0.30	0.30

**Table 2 toxins-14-00834-t002:** Changes in the gelatinization properties of corn treated by power ultrasound.

Treatment Time (min)	Peak Viscosity (mPa·s)	Through (mPa·s)	Breakdown (mPa·s)	Final Viscosity (mPa·s)	Setback (mPa·s)	Gelatinization Temperature (°C)
0	1422 ± 30a	1210 ± 31a	212 ± 10a	2390 ± 29a	1180 ± 2a	77.90 ± 0.49a
4	1578 ± 23b	1292 ± 16ab	256 ± 35a	2393 ± 45a	1151 ± 34a	77.58 ± 0.04a
6	1557 ± 35b	1316 ± 33b	252 ± 18a	2511 ± 30b	1195 ± 24a	77.53 ± 0.04a
8	1675 ± 33c	1396 ± 20bc	245 ± 36a	2581 ± 8b	1185 ± 12a	77.48 ± 0.04a
10	1648 ± 3c	1421 ± 19c	228 ± 16a	2560 ± 6b	1140 ± 25a	77.05 ± 0.57a
12	1715 ± 32c	1444 ± 20c	270 ± 30a	2598 ± 52b	1154 ± 11a	77.48 ± 0.04a

Values with different letters (a, b, c) were significantly different (*p* < 0.05).

**Table 3 toxins-14-00834-t003:** Changes in the thermodynamic properties of corn treated by power ultrasound.

Treatment Time (min)	T_0_ (°C)	T_p_ (°C)	T_c_ (°C)	T_c_–T_0_ (°C)	ΔH (J/g)
0	63.19 ± 0.14a	71.29 ± 0.24a	76.18 ± 0.16a	12.99 ± 0.16a	4.90 ± 0.08a
4	63.87 ± 0.13a	71.46 ± 0.22a	76.12 ± 0.11a	12.25 ± 0.13a	4.79 ± 0.08a
6	63.52 ± 0.21a	71.26 ± 0.37a	76.05 ± 0.20a	12.53 ± 0.18a	4.80 ± 0.28a
8	63.05 ± 0.14a	71.76 ± 0.13a	76.34 ± 0.18a	12.99 ± 0.24a	4.80 ± 0.18a
10	63.81 ± 0.23a	71.72 ± 0.34a	76.26 ± 0.11a	12.45 ± 0.34a	4.74 ± 0.16a
12	63.96 ± 0.11a	71.20 ± 0.26a	76.25 ± 0.14a	12.29 ± 0.24a	3.97 ± 0.35b

T_0_, onset temperature; T_p_, peak temperature; T_c_, conclusion temperature; T_c_–T_0_, transition temperature; Values with different letters (a, b) were significantly different (*p* < 0.05).

**Table 4 toxins-14-00834-t004:** Changes in the amylose content in corn treated with power ultrasound.

Treatment Time (min)	0	4	6	8	10	12
Amylose content (%)	29.80 ± 1.01a	30.05 ± 0.09ab	30.58 ± 0.75ab	32.09 ± 1.41ab	31.21 ± 0.13ab	32.93 ± 0.54b

Values with different letters (a, b) were significantly different (*p* < 0.05).

**Table 5 toxins-14-00834-t005:** Concentrations of AFB1 and ZEA standard solutions.

	AFB_1_ Concentration (μg/L)	ZEA Concentration (μg/L)
M1	2	2
M2	4	4
M3	8	8
M4	20	20
M5	40	40
M6	80	80
M7	160	160

AFB1, aflatoxin B1; ZEA, zearalenone.

**Table 6 toxins-14-00834-t006:** Confirmation results of the mycotoxin determination method.

	AFB1 in Corn	AFB1 in Soaking Water	ZEA in Corn	ZEA in Soaking Water
Linear range (μg/L)	2.5–160	2.5–160	2.5–80	2.5–160
Standard curve equations	Y = −85380.7 + 271689X	Y = −154813 + 245147X	Y = 9405.45 + 12111.4X	Y = −6128.01 + 9970.71X
Coefficient of determination (R^2^)	0.9963	0.9967	0.9981	0.9947
Limit of detection (μg/L)	0.72	0.70	2.12	2.72
Limit of quantification (μg/L)	2.43	2.10	7.06	8.16
Recovery (%)	85.8%	101.4%	90.3%	106.0%

AFB1, aflatoxin B1; ZEA, zearalenone.

## Data Availability

The datasets generated during and/or analysed during the current study are available from the corresponding author on reasonable request.

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
