# Peer review of "Reduction of Aflatoxin B1 and Zearalenone Contents in Corn Using Power Ultrasound and Its Effects on Corn Quality"

_toxins, 2022, doi:10.3390/toxins14120834_

Round 1
Reviewer 1 Report
The study is well designed and clearly presented, but the manufacturer names and model details of most equipment /instruments are missing. They should be added for reference.
Are the statistical analyses adequately referred to? At least they are not properly discussed.
Author Response
We thank you for your thoughtful suggestions and insights. The responses to the comments have been prepared in the attachment. Please see the attachment.

Reviewer 2 Report
This is an interesting manuscript describing the overall effects of ultrasound treatment on aflatoxin B1 and zearalenone in corn samples. The methodological approaches used are adequate, and the data reported are acceptable. This subject has been explored previously, but the study provided further evidences on the efects of ultrasound treatments on quality parameters of the matrix (in this case, corn). To improve the manuscript, the following items should be considered for revision:
-L.30-31: Please provide additional information on the toxicological properties of the mycotoxins evaluated.
-L.59-61: The degradation rates in Figure 1 should be explained in the caption. Also, the statistical inference must be displayed. Same regarding the other figures (2-6) in the manuscript.
-L.65-68: The decrease in degradation rates of AFB1 after 10 min. treatment should be further discussed, in addition to the possible transfer of the mycotoxin from water to corn powder, as the same pattern was observed for zearalenone. What about the possible interactions of mycotoxins with the food matrix under different ultrasound exposure times?
-L.171: Statistical inference is missing in Table 1.
-L.215: Data in Table 4 should be presented in the text, only (not in a table).
-L.245-246: This statement is somewhat contradictory with L.241-243.
-L.254-255: Naturally contaminated corn was used in the experiments? Please clarify. In this case, how the mycotoxins were determined?
Author Response

(The authors gave the same response as above.)

Reviewer 3 Report
This article aims to study the effects of power intensity, solid-liquid ratio, treatment time and modes of power ultrasound on the degradation rates of mycotoxins AFB1 and ZEA along with their effects on corn quality. Generally, this paper is methodically done and fairly written. However, the following points need to be addressed before this paper could be accepted for publication in Toxins.
1. Figures 1-3 can be combined to be parts of a single Figure 1.
2. Figures 4-6 can be combined to be parts of a single Figure 2.
3. Line 108-110 & Line 247-248 – here there is a correlation between power intensity and the ultrasonic treatment modes significantly affecting the degradation rates. However, in the conclusion section, it is mentioned as no significant correlation.
4. Table 4 – Do the authors think there is a significant difference in amylose contents with increasing ultrasound treatment time?
5. The scale bar for SEM images in Figure 8 is not clear or not provided.
6. All the purchase details of chemicals/reagents and instruments/equipment/kits should be provided as state, city and country in the case of USA as well as city and country in the case of other countries. Also, for the second instance of same vendor/company’s mention, the authors can simply mention the company name.
7. Materials and methods – provide a reference citation for procedure adopted in sections 4.2, 4.4, 4.5, 4.7, 4.8, 4.9.
8. Line 266 & 281 – “4000 r/min” should be “4000 rpm”?
9. Line 288 – UPLC column dimension should be (100 x 2.1 mm, particle size 1.7 mm).
10. Line 298, 299 – “aux gas” should be “auxiliary gas”
11. Table 6 – “limit of determination” should be “limit of detection”
12. Section 4.7 – the procedure is incomplete without the HPLC procedure.
13. Section 4.8 and 4.9 – the detailed procedure should be provided in the supplementary material.
14. Line 321 – is DSC defined at the first instance? Also please check with other abbreviations throughout the manuscript if the authors have defined them at the first instance and abbreviated thereafter.
15. Section 4.11 - What does “66.8” represent in the equation for amylose content determination?
16. Section 4.12 – the SEM sample preparation as well as the instrument details should be provided.
17. Section 4.13 - provide state, city and country name in the case of USA for statistical softwares SPSS and Origin.
18. Just before the conclusion, please include the diagrammatic representation of optimized values of power intensity, solid-liquid ratio and treatment modes as well as observed trends in amino acid, amylose, fatty acid and gelatinization, which should help the readers with a quick take home points.
Author Response

(The authors gave the same response as above.)

Round 2
Reviewer 3 Report
The authors have satisfactorily addressed all the comments raised by reviewers and therefore I recommend acceptance of this article for publication in Toxins.
Author Response

(The authors gave the same response as above.)
